# Oral Nutritional Supplementation in Older Adults with a Hip Fracture—Findings from a Bi-National Clinical Audit

**DOI:** 10.3390/healthcare12212157

**Published:** 2024-10-29

**Authors:** Jack J. Bell, Rebecca J. Mitchell, Ian A. Harris, Hannah Seymour, Elizabeth Armstrong, Roger Harris, Stewart Fleming, Sarah Hurring, Jacqueline Close

**Affiliations:** 1Principal Research Fellow, Allied Health Research Collaborative, The Prince Charles Hospital, Chermside, QLD 4032, Australia; 2Australian Institute of Health Innovation, Macquarie University, Level 6, 75 Talavera Road, Sydney, NSW 2109, Australia; r.mitchell@mq.edu.au; 3Ingham Institute for Applied Medical Research, School of Clinical Medicine, UNSW Medicine and Health, UNSW, Sydney, NSW 2170, Australia; ianharris@unsw.edu.au; 4Fiona Stanley Hospital, Perth, WA 6150, Australia; hannah.seymour@health.wa.gov.au; 5School of Population Health, UNSW, Sydney, NSW 2052, Australia; elizabeth.armstrong@unsw.edu.au; 6Australian and New Zealand Hip Fracture Registry Steering Group, Auckland 1010, New Zealand; rogerh14b@gmail.com; 7OperaIT Pty Ltd., 3994 Pacific Highway, Loganholme, QLD 4129, Australia; stewart@hipfracture.com.au; 8Te Whatu Ora, Waitaha Canterbury New Zealand, Christchurch 8011, New Zealand; sarah.hurring@cdhb.health.nz; 9Neuroscience Research Australia, Sydney, NSW 2031, Australia; j.close@neura.edu.au; 10School of Clinical Medicine, UNSW, Sydney, NSW 2052, Australia

**Keywords:** clinical audit, dietary supplements, frailty, hip fractures, hospitals, malnutrition, nutrition risk assessment, nutritional support

## Abstract

Background/Objectives: Evidence-based guidelines and care standards recommend offering oral nutrition supplements to all older adults with hip fracture, not just those already malnourished. This study aimed to identify the proportion of inpatients in a sample of hospitals in two countries that were provided with oral nutritional supplementation (ONS) following a hip fracture and to identify factors associated with ONS provision. Methods: An analysis of prospectively collected data from a bi-national Hip Fracture Registry nutrition sprint and registry audit data limited to older adults (≥65 years) undergoing surgical intervention for a fractured hip from 1 to 31 August 2021. Multivariable logistic regression was used to identify factors associated with providing ONS. Results: Patient-level data was available for 385 older adults (median 85 years; 60.5% female) admitted to twenty-nine hospitals. Less than half (*n* = 47.3%) of the audited inpatients were provided ONS. After adjusting for covariates, ONS was more likely to be provided to older adults who were identified as malnourished on formal testing (OR 11.92; 95%CI 6.57, 21.69). Other factors associated with prescription of ONS included those who did not have a preoperative medical assessment (OR 2.26; 95%CI 1.19, 4.27) or were cognitively impaired (OR 1.83; 95%CI 1.01, 3.32), severely frail, or terminally ill (OR 3.17; 95%CI 1.10, 9.17). Conclusions: ONS was provided in line with evidence-based recommendations for less than half of the older adults with a hip fracture in 29 hospitals in two countries. A structured approach to implementation may be required to reduce complications and improve outcomes for all older adults after a hip fracture, not just those assessed as cognitively impaired, frail, and/or malnourished.

## 1. Introduction

Malnutrition in older adults with hip fractures is a key predictor of morbidity, hospital-acquired complications, length of stay (LOS), supported living arrangements, 12-month mortality, and treatment costs [1,2,3,4]. International clinical audit data suggest that between 20 and 50% of acute hip fracture patients are malnourished on admission to hospitals [5].

Although exact recommendations vary globally, evidenced-based guidelines suggest energy intakes around 30 kJ/kg/day and high protein intakes of 1.2–1.5 g/kg/day, or even higher in the case of severe illness, injury, or malnutrition [6]. Achieving these intakes can be difficult in older adults with a hip fracture; consequently, oral nutritional supplements and multidimensional, multidisciplinary team interventions are recommended for all hip fracture patients to improve dietary intake, reduce complications, and improve outcomes [6,7,8]. Yet in many settings, provision of ONS and other nutrition care processes for older adults with hip fractures has been reliant on a positive malnutrition screen leading to referral and review by a dietitian [9]. Multiple factors highlight the need to consider all hip fracture patients “at-risk” of malnutrition. These include poor screening tool sensitivity across the most common tools resulting in underdiagnosis and treatment [10], mean protein and energy intakes below half of the recommended requirements in a hip fracture population leading to additional incident malnutrition [11], high negative impact of malnutrition on outcomes [1,2], and delayed access to dietetics care if reliant on dietitian-delivered nutrition care [12]. These, combined with the positive cost–benefit of early nutrition care [13], underpin recommendations for a proactive approach to multimodal, multidisciplinary intervention using oral nutritional supplementation (ONS) from the time of admission to the hospital [14]. Such an approach targets proactive enhancement of nutritional care and facilitates rehabilitation and recovery [15].

Whilst determinants of malnutrition in older adults with hip fractures are diverse, these collectively contribute to inadequate intake, increased requirements, and/or reduced nutrient bioavailability, leading to malnutrition [11,16]. Although there is no single solution to the complex problem of malnutrition, ONS is a routinely recommended component of multidisciplinary, multimodal care for all patients with a hip fracture [8,14,15]. Systematic reviews have concluded that ONS started before or soon after surgery appears well tolerated, improves protein and energy intakes to near-optimum levels, and may reduce LOS, risk of complications, and unfavourable outcomes, including death within the first 12 months after a hip fracture [3,8,15,17,18,19]. Guidelines from the European Society for Parenteral and Enteral Nutrition (ESPEN) recommend that all older patients with a hip fracture should be offered ONS regardless of their nutritional status after surgery (Grade A recommendation; 100% consensus) [6,20]. The recently updated Australian and New Zealand Hip Fracture Clinical Care Standard includes a focus on nutrition and a new performance indicator measuring the proportion of all admitted patients with a hip fracture who received protein and energy ONS during their admission [7].

The evidence and recommendations supporting proactive ONS for all hip fracture patients, not just those overtly malnourished, are clear. We hypothesised that variation exists within and between Australian and New Zealand hospitals in the provision of oral nutritional supplements to older adults with hip fractures. This study aimed to identify the proportion of inpatients across Australian and New Zealand hospitals who were provided or prescribed ONS following a hip fracture and to identify factors associated with ONS.

## 2. Materials and Methods

### 2.1. Study Design

A cohort study aligned with the STROBE guidelines was undertaken combining data from the Australian and New Zealand Hip Fracture Registry (ANZHFR) and ANZHFR Nutrition Sprint Audit.

### 2.2. Setting

The ANZHFR is an established clinical quality registry that reports on processes and outcomes of hip fracture care with the intent that data are used to drive local quality improvement activity. In 2021, the ANZHFR collected individual patient data for 15,331 patients across 86 participating hospitals in Australia and New Zealand.

In 2021, the ANZHFR conducted a bi-national sprint audit, the ANZHFR Nutrition Sprint Audit. A Sprint Audit involves collecting a small number of additional topic-specific variables over an agreed-upon timeframe with a view to identifying potential gaps in care and opportunities to improve care. Hospitals elect to opt-in to Sprint Audits and collect these additional variables. The Nutrition Sprint Audit (NSA) was designed to capture information on nutrition-specific hip fracture care. The data variables were added to the minimum data set for a one-month period and collected for each eligible person admitted to a hospital with a hip fracture within the defined period. This “snapshot” or “sprint” approach reduces burden on sites while still providing useful clinical information that can be used to improve care.

The sprint protocol and dataset definitions were developed by the authors with consideration to the national and international peer-reviewed literature [10,12,14,17], practice recommendations [20,21], and guidelines [22,23] and piloted in three sites.

### 2.3. Participants

A convenience sample of sites opted-in to participate in the sprint audit. Recruitment of sites was supported through ANZHFR newsletters and direct communication with sites. Sprint patient eligibility criteria were as follows: persons aged 50 years and older admitted to an ANZHFR participating hospital with a fractured hip from a minimal or low trauma injury. Exclusion criteria specific to this study were as follows: patients aged less than 65 and those with missing data. ANZHFR and NSA data were collected and entered by team members in local sites in line with ANZHFR audit processes. The NSA took place over a 1-month period between 1 August and 31 August 2021.

### 2.4. Variables

The primary outcome was whether ONS was provided or not (Yes/No). ONS was defined as a form of oral nutrition support recommended when oral intake is expected to be inadequate to improve dietary intake and reduce the risk of complications [20,24]. This definition included both “pharmaceutical” oral nutrition supplements prescribed in the medication chart or ONS that were individually evidenced, for example, through documentation in medical records or foodservice ordering systems. Malnutrition was defined at local sites using tools validated for the purpose of identifying protein/energy malnutrition. Across Australia and New Zealand, this is most commonly the Subjective Global Assessment or International Classification of Diseases criteria [25,26,27]. Other covariates of interest were those available in the data and known predictors from the literature and included jurisdiction of admission (Australia or New Zealand), age group, sex, place of residence pre-admission, ward type, pre-admission cognitive status, frailty status, pre-operative medical assessment, geriatric assessment, American Society of Anaesthesiologists (ASA) physical status classification score, and delirium assessment [28].

### 2.5. Approvals

Pre-collected data from the Australian and New Zealand Hip Fracture Registry (ANZHFR) was utilised for this study. Ethics approval, including explicit approvals for the nutrition sprint audit, was granted for each jurisdiction as follows, with the exception of Queensland, where the requirements of state-specific legislation precluded the contribution of patient-level data in the timeframe required for a sprint audit: NSW: NSW Population Health Services Research Ethics Committee, reference HREC/14/CIPHS/51; VIC: Monash Health HREC, reference HREC/16/MONH/65; QLD: The Prince Charles Hospital HREC, reference HREC/14/QPCH/54; WA: Sir Charles Gairdner Group HREC, reference 2014-043; SA: Central Adelaide Local Health Network HREC, reference HREC/14/RAH/115; NT: Health and Menzies School of Health Research HREC, reference 2023-4526; TAS: UTAS HREC, reference H0015534 and H0017654; New Zealand: Northern B Health and Disability Ethics Committee, reference 14/NTB/112; UnitingCare Health HREC, reference 2003; Mercy Health HREC, reference 2020-053; St John of God Health HREC, reference 1853; Ramsay Health Care WA|SA HREC, reference 1647; Australian Institute of Health and Welfare HREC, reference EO2019-3-1065. A waiver of consent is used in New South Wales, Queensland, and South Australia for the collection of the clinical registry data. An opt-out consent is used in Tasmania, Western Australia, Victoria, and New Zealand.

### 2.6. Statistical Methods

Data were analysed using the SPSS Statistical Analysis Software Version 28.0.1.0 (©IBM Corp (Armonk, NY, USA) and its licensors, 1989, 2021). Data were assessed for normality and non-parametric approaches applied where indicated. Descriptive statistics included mean (standard deviation) or median (interquartile range). Categorical data were reported as counts and percentages, with basic comparisons made using Pearson Chi Square statistics.

Univariate and multivariable logistic regressions were used to examine factors associated with receiving ONS. Predictor variables were included in multivariable analysis where those that significantly contributed to the likelihood of receipt of ONS at a *p*-value of <0.25 during a univariate analysis or were considered clinically relevant regardless of significance (age group, sex), and where they did not demonstrate collinearity with other predictor variables (tolerance < 0.2, VIF ≥ 5). Odds ratios and 95% confidence intervals (CI) were calculated for univariate and multivariate analyses. For parsimony and to minimise potential problems with model convergence resulting from smaller sample sizes, audit variables were collapsed clinically and statistically relevant to achieve the recommended sample size for logistic regression. For all other tests, a predefined *p*-value of <0.05 was considered statistically significant.

## 3. Results

Twenty-four Australian and five New Zealand hospitals participated in the Sprint Audit from a potential pool of 93 hospitals (Figure 1). The initial sample comprised 450 patients, but this was reduced to a final sample of 385 patients after excluding patients with missing data from the ANZHFR dataset (*n* = 29), missing ONS data (*n* = 16), and patients <65 years (*n* = 20).

Patients were mostly older (median age 85 years; interquartile range—12 years) and female (60.5%). Almost one in three patients was diagnosed as malnourished (30.9%). ONS was provided or prescribed for less than half of all older adults undergoing surgical intervention for hip fracture (*n* = 182, 47.3%) (Table 1).

Table 1 highlights the characteristics of patients who were or were not provided ONS. Country of admission, pre-admission cognitive status, frailty, delirium, malnutrition, and discharge destination were significantly associated with the provision of ONS.

Figure 2 illustrates the variability in the proportion of patients provided ONS across participating jurisdictions.

Univariate and multivariable predictors of oral nutrition supplement provision and/or prescription are provided in Table 2. Impaired cognition or known dementia, delirium, frailty, and malnutrition were significantly associated with receiving ONS in univariate analysis. Country of admission, pre-operative medical assessment, and ASA score were also statistically considered appropriate for inclusion in the multi-variable model (*p* < 0.25). Although age and sex were not considered statistically significant, these variables were included in multivariable analysis as clinically relevant factors. After adjusting for these factors, multivariable regression demonstrated that patients who were known to be cognitively impaired prior to admission (OR 1.83; 95%CI 1.01, 3.32), did not receive a pre-operative medical assessment (OR 2.26; 95%CI 1.19, 4.27), who were severely frail or terminally ill (OR 3.17; 95%CI 1.10, 9.17), and/or who had a positive malnourishment assessment (OR 11.92; 95%CI 6.57, 21.69) had a higher likelihood of receiving ONS. Patients who were admitted to New Zealand hospitals were significantly less likely to receive oral nutrition supplements (OR 0.45; 95%CI 0.21, 0.96) after adjusting for clinically and statistically relevant factors.

## 4. Discussion

Evidence-based guidelines and care standards recommend that all patients with a hip fracture should be offered oral nutrition supplements, not just those obviously at risk or already malnourished [6,7,20]. To the authors knowledge, this is the first large-scale study to report the substantial gap between these recommendations and clinical practice and the factors associated with the provision of ONS. This study is also the first study globally to highlight the added benefit of nutrition sprint audits as an adjunct to core hip fracture clinical registry datasets to clearly articulate the variation between recommendations for oral nutrition supplementation and practice across settings.

The audit showed that ONS is being directed towards those most nutritionally vulnerable, for example, those with cognitive impairment, severe frailty, and/or already diagnosed malnutrition. However, over the last decade, studies have demonstrated the need to include ONS as part of multi-component interventions for all hip fracture patients after surgical intervention in order to improve intake, reduce the risk of complications, and improve healthcare outcomes [6,7,8,11,14,15,17,18,20]. This is particularly important given the poor sensitivity of nutrition screening tools commonly used and the inadequate attention to the considerable adverse impact of a co-diagnosis of overweight or obesity and protein-energy malnutrition on the outcomes after hip fracture [10,29].

Our study demonstrates that less than half of older adults received ONS after hip fracture surgery. This is less than that routinely reported in randomised controlled trials in hip fractures, with a recent meta-analysis describing adherence rates between 64.7 and 100% [8,17]. Although reasons for these differences are not well described, we suggest that tightly controlled, efficacy-focused studies may over-represent adherence to therapy in day-to-day practice [30]. Surprisingly, those who did *not* receive a preoperative assessment were much more likely to receive supplements. This may suggest discretionary decision-making by clinicians focusing on the highest-risk individuals, such as those with cognitive impairment, frailty, or malnutrition. It is also possible that sites that do not have routine preoperative medical assessment have protocols and pathways to support utilisation of ONS in hip fracture settings [31,32]. We suggest consideration of systematised approaches, for example, using pathways or protocols to routinely align ONS with national recommendations, reduce the incidence of hospital-acquired malnutrition and other nutrition-related complications, and improve patient and healthcare outcomes [6,7,20,24].

The higher rates of ONS in Australia compared with New Zealand may be explained by financial penalties for hospital-acquired malnutrition, case-mix incentivisation for diagnosis of malnutrition, and national accreditation standards (which specifically include nutrition care) in Australia [10,24,33]. The inclusion of an ONS indicator in the recently revised Hip Fracture Clinical Care Standard is expected to further improve the proportion of hip fracture patients provided with ONS during their hospital admission [7]. Whilst there is no direct financial incentive for ONS provision in either Australia or New Zealand, the consideration of nutrition as part of a best practice tariff or other incentivisation structure may be an opportunity for future works to improve outcomes [34].

Within Australia, there is marked variation in practice between the States (Figure 2). Australia has a multi-tiered healthcare system with national, state, and local funding and governance structures. It is unclear whether the observed variation reflects differences in state-level drivers or the nature of the hospitals participating in the audit. Anecdotally, lack of decision-maker awareness or trust in the existing evidence demonstrating the cost–benefit of nutrition support and recommendations for the routinisation of ONS have been observed [3,6,13,15,18,35]. However, it is also likely that variation is at a hospital level, influenced by individual (e.g., nutrition care champions) and institutional (e.g., local policies or care pathways, information technology infrastructure, nutrition culture) factors [36,37,38,39].

This study demonstrates the utility of sprint audit data as a supplementary tool to enhance the function of clinical quality registries to identify variance in care that is delivered according to standards [7]. Opportunities exist in Australia and New Zealand to improve the provision of ONS in a hip fracture population. Having identified gaps in care, it is essential that we move forward to address those gaps by including exploratory analysis to identify barriers and enablers, and then develop, implement, and evaluate interventions (behaviour change techniques and mode(s) of delivery) that are underpinned by relevant implementation science theories, models, and frameworks [40,41,42,43]. Ideally, these will be undertaken as a cluster randomised implementation effectiveness hybrid trial [42,44].

Although audit findings are inclusive of patients across 29 Australian and New Zealand hospitals, a key limitation is that the study sample size by jurisdiction or site was not adequate to support statistical comparisons at the state or hospital level. Given that only 29 of 93 registry sites were included in the final sample, we cannot exclude the possibility of selection bias, which may affect generalisability. Finally, the pragmatic nature of the audit precluded auditing adherence to the provided ONS.

## 5. Conclusions

Evidence-based guidelines, care standards, and registry audit datasets support routinely offering oral nutritional supplements to all older adults with a hip fracture, not just those obviously at risk. Less than half of older adult inpatients with a hip fracture audited in a sample of 29 hospitals across Australia and New Zealand received ONS. Provision and prescription are targeted to those who are cognitively impaired, extremely frail, or already malnourished. The next logical step is to undertake a structured implementation science approach to close the evidence-to-practice gap, with the ultimate aim of improving outcomes for all adults following a hip fracture.

## Figures and Tables

**Figure 1 healthcare-12-02157-f001:**
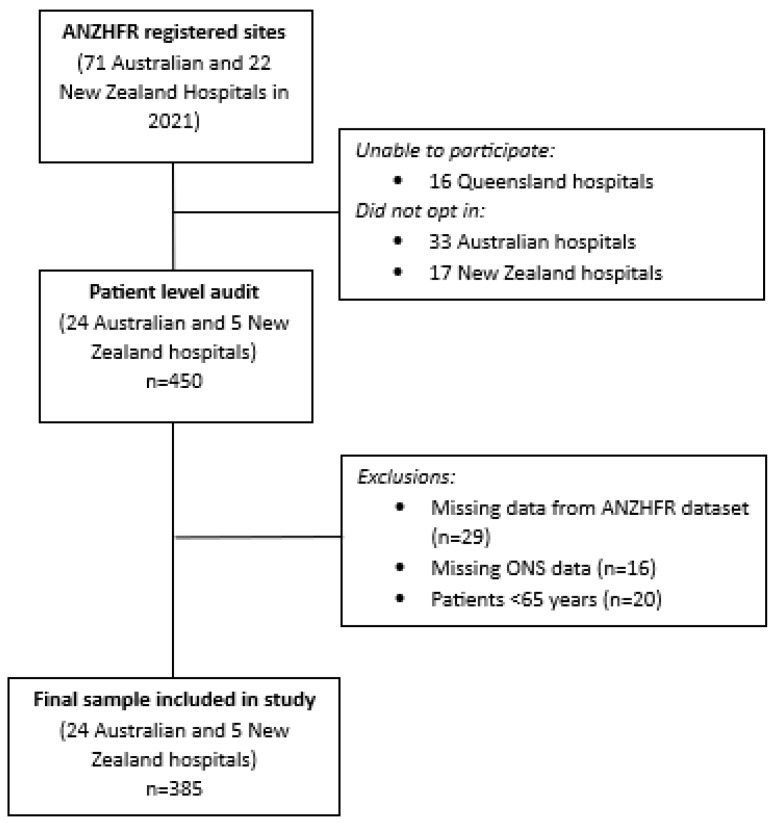
Participant flow diagram.

**Figure 2 healthcare-12-02157-f002:**
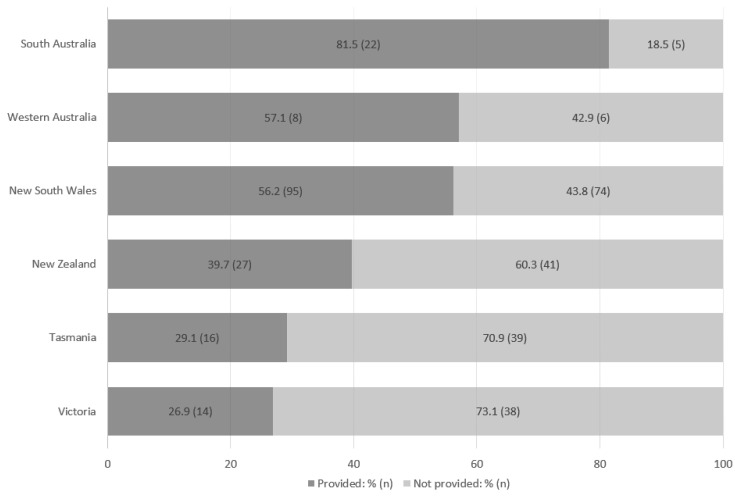
Oral nutritional supplementation provided to patients by jurisdiction.

**Table 1 healthcare-12-02157-t001:** Characteristics of participants who were or were not provided oral nutrition supplements.

Characteristics *n* (%)	ONS not Provided *n* = 203 (52.7%)	ONS Provided *n* = 182 (47.3%)	*p*-Value ^5^
Country of admission:			
New Zealand	41 (20.2)	27 (14.8)	Χ^2^(1) = 1.90,
Australia	162 (79.8)	155 (85.2)	*p* = 0.168
Age:			
65–79	66 (32.5)	55 (30.2 )	Χ^2^(1) = 0.23
≥80	137 (67.5)	127 (69.8)	*p* = 0.629
Sex:			
Female	121 (40.4)	112 (38.5)	Χ^2^(1) = 0.15, *p* = 0.699
Male	82 (59.6)	70 (61.5)	
Usual place of residence ^1^			
Private residence	154 (76.6)	137 (75.3)	Χ^2^(1) = 0.09, *p* = 0.759
Residential aged care facility	47 (23.4)	45 (24.7)	
Ward type ^1^			
Hip fracture unit/orthopaedic ward/preferred ward	181 (90.5)	160 (88.9)	Χ^2^(1) = 0.69, *p* = 0.406
Outlying ward, HDU, ICU or CCU	19 (9.5)	20 (11.1)	
Pre-admission cognitive status ^1^			
Normal cognition	145 (71.8)	93 (52.2)	Χ^2^(1) = 15.43, *p* < 0.001
Impaired cognition/known dementia	57 (28.2)	85 (47.8)	
Pre-operative medical assessment ^1,2^			
No assessment conducted	39 (19.5)	49 (27.2)	Χ^2^(1) = 3.18, *p* = 0.075
Assessment conducted	161 (80.5)	131 (72.8)	
Assessment by geriatric medicine ^1^			
No	20 (9.9)	10 (5.5)	Χ^2^(2) = 2.88, *p* = 0.236
Yes	169 (83.7)	157 (86.3)	
No service	13 (6.4)	15 (8.2)	
ASA ^3^			
Grade I or II	38 (18.7)	23 (12.6)	Χ^2^(3) = 4.42, *p* = 0.219
Grade III	116 (57.1)	101 (55.5)	
Grade IV or V	30 (14.8)	34 (18.7)	
Not known	19 (9.4)	24 (13.2)	
Clinical frailty scale ^4^			
Not frail/not at risk	49 (24.1)	15 (8.2)	Χ^2^(4) = 23.62, *p* < 0.001
Apparently vulnerable	21 (10.3)	17 (9.3)	
Mild/moderate frailty	69 (34.0)	65 (35.7)	
Severe frailty	18 (8.9)	36 (19.8)	
Not known	46 (22.7)	49 (26.9)	
Delirium assessment			
Assessed and not identified	94 (46.3)	75 (41.2)	Χ^2^(2) = 12.94, *p* = 0.002
Assessed and identified	49 (24.1)	73 (40.1)	
Not known	60 (29.6)	34 (18.7)	
Malnutrition assessment			
Not malnourished	156 (76.8)	77 (42.3)	Χ^2^(2) = 91.72, *p* < 0.001
Malnourished	20 (9.9)	99 (54.4)	
Not known	27 (13.3)	6 (3.3)	

^1^ Missing < 10 cases. ^2^ Preoperative medical assessment by geriatrician or team member, physician or team member, GP, or specialist nurse. ^3^ Merged ASA Physical Status Classification System (ASA) scores: I–II, III, IV–V. ^4^ Categorised clinical frailty scale scores: not at risk (1–3), vulnerable (4), mild/moderate (5–6), severe (7+), and not known. HDU: high-dependency unit; ICU: Intensive Care Unit; CCU: Coronary Care Unit. ^5^ Pearson Chi-Square statistic.

**Table 2 healthcare-12-02157-t002:** Univariate and multivariable predictors of oral nutrition supplement provision and/or prescription.

	Univariate		Multivariable	
	OR (95% CI)	*p*-Value	OR (95% CI)	*p*-Value
**Country of admission**				
Australia	1		1	-
New Zealand	0.69 (0.40, 1.17)	*p* = 0.170	0.45 (0.21, 0.96)	*p* = 0.039
**Age group ^1^**				
65–79	1		1	-
≥80	1.11 (0.72, 1.71)	*p* = 0.629	0.72 (0.42, 1.23)	*p* = 0.224
Sex:				
Female	1		1	-
Male	0.92 (0.61, 1.39)	*p* = 0.699	1.15 (0.69, 1.90)	*p* = 0.591
**Usual place of residence**				
Private residence ^2^	1		-	
Residential aged care facility	1.09 (0.68, 1.74)	*p* = 0.718	-	-
**Pre-admission cognitive status**				
Normal cognition ^3^	1		1	-
Impaired cognition or known dementia	2.25 (1.47, 3.43)	*p* < 0.001	1.83 (1.01, 3.32)	*p* = 0.045
**Admission ward type**				
Preferred ward ^4^	1		-	-
Outlying ward or HDU/ICU/CCU	1.20 (0.62, 2.32)	*p* = 0.597	-	-
**Preoperative medical assessment conducted**				
Yes ^5^	1	-	1	-
No	1.55 (0.96, 2.50)	*p* = 0.073	2.26 (1.19, 4.27)	*p* = 0.013
**Assessed by geriatric medicine**				
Yes ^6^	1		-	-
No/no service	0.82 (0.47, 1.44)	*p* = 0.491	-	-
**ASA score**				
Grade I or II	1		1	
Grade III	1.44 (0.80, 2.58)	*p* = 0.221	0.98 (0.47, 2.03)	*p* = 0.954
Grade IV or V	1.87 (0.92, 3.82)	*p* = 0.085	0.93 (0.37, 2.37)	*p* = 0.884
Not known	2.09 (0.94, 4.62)	*p* = 0.069	2.13 (0.80, 5.54)	*p* = 0.129
**Delirium assessment**				
Assessed and not identified	1		1	
Assessed and identified	1.87 (1.16, 3.00)	*p* = 0.010	0.92 (0.49, 1.73)	*p* = 0.792
Not known/not assessed	0.71 (0.42, 1.19)	*p* = 0.196	0.86 (0.45, 1.64)	*p* = 0.644
**Clinical frailty scale**				
Very fit, well, or well with comorbid disease	1		1	
Apparently vulnerable	2.64 (1.12, 6.26)	*p* = 0.027	2.37 (0.86, 6.50)	*p* = 0.094
Mildly or moderately frail	3.08 (1.57, 6.02)	*p* = 0.001	1.95 (0.84, 4.52)	*p* = 0.121
(Very) severely frail or terminally ill	6.53 (2.91, 14.67)	*p* < 0.001	3.17 (1.10, 9.17)	*p* = 0.033
**Malnutrition assessment**				
Not malnourished or not known	1	-	1	-
Malnourished	10.91 (6.32, 18.84)	*p* < 0.001	11.92 (6.57, 21.69)	*p* < 0.001

^1^ Age in years was categorised in line with MeSH categories for aged 65–79 years and aged 80 and over. ^2^ Including other (*n* = 1), not known (*n* = 2). ^3^ Including not known (*n* = 5). ^4^ Hip fracture unit, orthopaedic unit, or other preferred ward, including not known (*n* = 5). ^5^ Pre-operative medical assessment by geriatrician or team member, physician or team member, GP, or specialist nurse, including not known (*n* = 5). ^6^ Including not known (*n* = 1).

## Data Availability

Information about accessing ANZHFR data is available from the ANZHFR registry website: https://anzhfr.org/data-access/ (accessed on 26 May 2024).

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
