# Peer review of "Oral Nutritional Supplementation in Older Adults with a Hip Fracture—Findings from a Bi-National Clinical Audit"

_healthcare, 2024, doi:10.3390/healthcare12212157_

Round 1
Reviewer 1 Report
Comments and Suggestions for Authors
Are we providing oral nutrition supplements in line with recommendations for older adults with a hip fracture? Findings from a bi-national clinical audit
· Title is misleading and needs to be revised according to the content of article, please make it simple, attractive and innovative.
· It would be better to organize the keywords in alphabetical order and keep the repeated words only.
· Introduction is well organized and written in systematic way, however subsequent sections lack story line.
· Authors have not explained all the parameters in the introduction section, please give some dietary recommendations according to RDA, WHO, FAO or any other with their references, furthermore authors should explain different studies as an example with a hip fracture in adults.
· Authors should briefly explain about the rationale and reasoning of the study in the end of introduction section with special focus on why the study was conducted what could be the outcomes etc.
· In line 81 and 82 either abbreviations are accepted or self-made please check and clear.
· Please give some previous values and clinical studies to highlight the importance of the above said topic.
· Please provide comprehensive inclusion and exclusion criteria with complete details to enroll the participant in this study.
· Provide a consent form that was used to get consent from the participants.
· Also provide consort form mentioning all the groups and their participants.
· Results needs some more clarity and statistical explanation.
· Authors should properly explain all the results and discussion and compare with different studies also mention the reasoning in discussion section.
Comments on the Quality of English Language
Moderate editing of English language required.
Author Response
Please refer attached response to reviewer 1 feedback.

Reviewer 2 Report
Comments and Suggestions for Authors
This outstanding article examines the use of Oral Nutritional Supplements (ONS) in hip fracture patients of any kind. The set-up and work-up are very nicely done. I do have some comments:
Statistics: Why did you choose the Bernouilli regression over the Poisson regression?
Tables: is there any information available on the patients' education and/or retirement income?
Author Response
Please find attached response to review 2 feedback.

Reviewer 3 Report
Comments and Suggestions for Authors
I would like to thank the Editor for the invitation to review this article. I commend the authors for the quality of the manuscript and for addressing a very interesting topic. However, I would suggest a few minor revisions:
- Introduction: The text is well-written, clear, and flows smoothly.
- Methods: While the methodological content is presented appropriately, I recommend dividing the Methods section into subsections to improve readability and avoid a single, dense block of text.
- Methods: The manuscript does not include a section regarding ethical approval for the study, which should be added.
- Results: The tables and figures are comprehensive and well-structured, with adequately described legends. However, I find the Results text somewhat brief and suggest elaborating on the findings.
- Discussion/Conclusions: The discussion and conclusions are well-written and clear.
- References: Many citations are outdated. Would it be possible to include more recent references to strengthen the manuscript?
- Please ensure to complete the following section: Institutional Review Board Statement.
Specifically,
Methods 1. The authors state that a cohort study was conducted. To this end, the Methods section should indicate that the study was reported in accordance with the STROBE guidelines, and the corresponding checklist should be completed. https://www.equator-network.org/reporting-guidelines/strobe/ Results 1. The sample recruitment in the initial section of the results is adequately described. Would it be possible to include a flow chart as an image to visually depict the recruitment process? 2. Based on the variables described in Table 1 and Table 2, it would be helpful to provide a more comprehensive description of the results, not just focusing on the significant findings. 3. Table 2 is missing a title; please add one. Discussion 1. Lines 193-202: The study results could benefit from being discussed in comparison with other international data to better support the findings. 2. Line 209: It is mentioned that in Australia, there is a difference in practice between states. This point could be further clarified, as a European reader might find it challenging to understand. 3. Line 217: Could you provide further clarification on this section by adding the future perspectives offered by your study? I also suggest incorporating these aspects into the conclusions to create a more structured closing statement, replacing "structured implementation approach."Author Response
Please find attached response to reviewer 3 feedback.

Reviewer 4 Report
Comments and Suggestions for Authors
This is a convincing study and manuscript.
However, the authors may further elaborate on the reasons behind malnutrition among older adults with hip fracture in the background section.
Furthermore, you should elaborate on how the hospitals taking part in the study have been selected/recruited.
Author Response
Please find attached response to reviewer 4 feedback
